# Artificial Intelligence Applied to Electrical and Non-Invasive Hemodynamic Markers in Elderly Decompensated Chronic Heart Failure Patients

**DOI:** 10.3390/biomedicines12040716

**Published:** 2024-03-22

**Authors:** Gianfranco Piccirillo, Federica Moscucci, Martina Mezzadri, Cristina Caltabiano, Giovanni Cisaria, Guendalina Vizza, Valerio De Santis, Marco Giuffrè, Sara Stefano, Claudia Scinicariello, Myriam Carnovale, Andrea Corrao, Ilaria Lospinuso, Susanna Sciomer, Pietro Rossi

**Affiliations:** 1Department of Internal and Clinical Medicine, Anesthesiology and Cardiovascular Sciences, Policlinico Umberto I, “Sapienza” University of Rome, 00185 Rome, Italy; gianfranco.piccirillo@uniroma1.it (G.P.); martina.mezzadri@uniroma1.it (M.M.); cristina.caltabiano@uniroma1.it (C.C.); giovanni.cisaria@uniroma1.it (G.C.); guendalina.vizza@uniroma1.it (G.V.); valerio.desantis@uniroma1.it (V.D.S.); marcogiuffre25@gmail.com (M.G.); sara.stefano@uniroma1.it (S.S.); claudia.scinicariello@uniroma1.it (C.S.); myriam.carnovale@uniroma1.it (M.C.); andrea.corrao@uniroma1.it (A.C.); susanna.sciomer@uniroma1.it (S.S.); 2Department of Internal Medicine and Medical Specialties, Policlinico Umberto I, Viale del Policlinico, 155, 00161 Rome, Italy; lospinuso.i@gmail.com; 3Arrhythmology Unit, Fatebenefratelli Hospital, Isola Tiberina-Gemelli Isola, 00186 Rome, Italy; rossi.ptr@gmail.com

**Keywords:** advanced heart failure, bioimpedance cardiography, QT, Tpeak-Tend, QT variability, temporal dispersion of repolarization phase, mortality

## Abstract

Objectives: The first aim of this study was to assess the predictive power of Tend interval (Te) and non-invasive hemodynamic markers, based on bioimpedance in decompensated chronic heart failure (CHF). The second one was to verify the possible differences in repolarization and hemodynamic data between CHF patients grouped by level of left ventricular ejection fraction (LVEF). Finally, we wanted to check if repolarization and hemodynamic data changed with clinical improvement or worsening in CHF patients. Methods: Two hundred and forty-three decompensated CHF patients were studied by 5 min ECG recordings to determine the mean and standard deviation (Te_SD_) of Te (first study). In a subgroup of 129 patients (second study), non-invasive hemodynamic and repolarization data were recorded for further evaluation. Results: Total in-hospital and cardiovascular mortality rates were respectively 19 and 9%. Te was higher in the deceased than in surviving subjects (Te: 120 ± 28 vs. 100 ± 25 ms) and multivariable logistic regression analysis reported that Te was related to an increase of total (χ^2^: 35.45, odds ratio: 1.03, 95% confidence limit: 1.02–1.05, *p* < 0.001) and cardiovascular mortality (χ^2^: 32.58, odds ratio: 1.04, 95% confidence limit: 1.02–1.06, *p* < 0.001). Subjects with heart failure with reduced ejection fraction (HFrEF) reported higher levels of repolarization and lower non-invasive systolic hemodynamic data in comparison to those with preserved ejection fraction (HFpEF). In the subgroup, patients with the NT-proBNP reduction after therapy showed a lower rate of Te, heart rate, blood pressures, contractility index, and left ventricular ejection time in comparison with the patients without NT-proBNP reduction. Conclusion: Electrical signals from ECG and bioimpedance were capable of monitoring the patients with advanced decompensated CHF. These simple, inexpensive, non-invasive, easily repeatable, and transmissible markers could represent a tool to remotely monitor and to intercept the possible worsening of these patients early by machine learning and artificial intelligence tools.

## 1. Introduction

Despite a series of innovative drug therapies (sacubitril, SGLT2 inhibitors, etc.) and medical devices (ICD and CRT), decompensated chronic heart failure (CHF) remains one of the major causes of mortality, morbidity, and rehospitalization especially in developed countries [1,2,3]. SARS-CoV-2 resulting in coronavirus disease (COVID-19) and other infections requiring medical attention have highlighted the excessive use of hospitalization. To avoid recurrent hospitalizations, it is necessary to introduce non-invasive, simple, inexpensive, transmissible, and early markers of CHF decompensation [4,5,6,7,8], especially since each hospitalization for acute decompensated heart failure increases the risk of worsening clinical conditions and mortality [9,10,11]. 

The European Society of Cardiology [12], the American Heart Association, the American College of Cardiology, and the Heart Failure Society of America [13] have introduced a new patient category with CHF, who have heart failure with an intermediate level of ejection fraction (see Methods), to better assess the cardiovascular risk of mortality and possible appropriate new drug therapy. Therefore, the first aim of this study was to evaluate non-invasive ECG repolarization-based markers based on this new classification. In particular, it has been previously observed that an increase in the Tpeak–Tend (Te) interval was a risk factor for total and cardiovascular mortality [14,15,16]. An increase in this interval and its short-period temporal dispersion was associated with mortality and decompensation in patients with acute CHF [17,18,19,20,21,22,23]. Finally, non-invasive hemodynamic parameters were studied based on bioimpedance to assess the possibility of monitoring clinical improvements in these patients [19,24,25,26,27]. The final aim of this work is to identify the electrical, bioimpedance, and biohumoral variables which, through artificial intelligence and machine learning devices, will be able to predict the clinical exacerbation of heart failure in the preliminary phases. By intercepting it early, it will be possible to treat the acutely decompensated CHF in the very early stages, trying to avoid hospitalization and the health and non-medical costs that would arise from it.

## 2. Methods

### 2.1. Study Subjects and ECG Analysis

We enrolled 243 consecutive patients admitted to our Geriatric Department from January 2019 to October 2023 due to dyspnea or other classic symptoms of decompensated chronic heart failure (CHF) according to the 2021 European Society of Cardiology heart failure guidelines [7] (Figure 1). 

Each patient enrolled disclosed medical history and the following assessments were performed: physical examination, standard ECG, transthoracic echocardiogram, and a 5 min of single-lead (II lead) ECG recording. Subsequently, we grouped patients according to left ventricular ejection fraction detected by echocardiography (LVEF_Ech_) [12], decompensated heart failure with reduced ejection fraction (HFrEF) with LVEF_Ech_ ≤ 40%, those with mildly reduced ejection fraction (HFmrEF) with LVEF_Ech_ levels of 41–49%, and, finally, those with preserved ejection fraction (HFpEF) with LVEF_Ech_ ≥ 50% [12]. All patients with CHF had stable previous clinical conditions at home with NYHA class II–III and all of them were in the IV NYHA functional class at the time of enrollment. Furthermore, at this time, we collected blood samples for quantifying NT-proBNP levels (Alere Triage Analyzer, Alere, San Diego, CA, USA) and other clinical chemistry analysis. ECG signals were acquired and digitalized with a custom-designed card (National Instruments, Austin, TX, USA) at a sampling frequency of 500 Hz. Measurements used for the ECG segment and interval analysis were detected automatically by a classic adaptive first derivative/threshold algorithm and a template method [28,29]. Our research group designed and produced software for data acquisition, storage, and analysis with the LabView program (National Instruments, Austin, TX, USA). An expert cardiologist (GP) checked the different ECG intervals and segments, automatically marked by the software, and manually corrected the mistakes as needed [28,29,30,31,32,33].

The means and standard deviations of QT intervals were obtained, calculated from q to end of T wave (QTe and QTe_SD_), QTpeak intervals (QTp and QTp_SD_), and obtained from q to peak of T wave. Finally, we calculated the mean and standard deviation (Te_SD_) of Te intervals [11,15,16,19,20,23,24,25,34,35]. 

One hundred and twenty-nine of the originally enrolled patients underwent non-invasive hemodynamic evaluation by means of bioimpedance cardiography (PhysioFlow; Manatec Biomedical, Poissy, France). This assessment was performed upon hospital admission and discharge (second study) (Figure 1) [19]. 

Upon patient discharge, a second 5 min ECG for repolarization variables and a second NT-proBNP test were administered. Therefore, the second study evaluated the following data parameters at the beginning of hospitalization and at discharge: hemodynamic, repolarization, and NT-proBNP data. The non-invasive hemodynamic data included the following: heart rate (HR), stroke volume (SV), stroke volume index (SVI), cardiac output (CO), cardiac index (CI), systemic vascular resistance (SVR), systemic vascular resistance index (SVRI), left ventricular ejection fraction (LVEF_BIO_), contractility index (ConI), left ventricular ejection time (LVET), cardiac work index (CWI), left ventricular end-diastolic volume (LVEDV), and early diastolic filling ratio (EDFR) [7,18,19,33,36,37,38,39,40,41,42]. Finally, upon hospitalization discharge, both the 5 min ECG and non-invasive hemodynamic recording were repeated. 

The study obtained formal approbation by the Ethical Committee of Policlinico Umberto I. The study code was BNP_HRV01 and the protocol number of the approbation document was 282/2023. Moreover, the study was first online registered on ClinicalTrials.gov as NCT04127162.

### 2.2. Statistical Analysis 

All variables with a normal distribution were expressed as mean ± standard deviation, whereas non-normally distributed variables were expressed as a median and inter-quartile range (i.r.), and categorical variables were expressed as frequencies and percentages (%). First, we compared the data obtained from three groups of CHF with different LVEF_Ech_ in the first study. More specifically, categorical variables were analyzed with the χ^2^ test. On the contrary, one-way ANOVA and Bonferroni tests were used to compare data for the normally distributed variables, while the Kruskal–Wallis and Mann–Whitney tests were used to compare non-normally distributed variables (as evaluated by Kolmogorov–Smirnov test). 

Secondly, we compared the ECG, non-invasive hemodynamic, and clinical data between deceased and surviving patients and between responders or non-responders to the drug therapy. In the second study, responders were defined as patients with a reduction of NT-proBNP at the end of hospitalization and non-responders were defined as patients without a reduction of NT-proBNP. In particular, a Student’s *t*-test and Mann–Whitney test were used respectively for the normally and non-normally distributed variables. Finally, a paired *t*-test and Wilcoxon test were used respectively to compare normally and non-normally distributed data at the beginning and at the end of the study. Multivariable forward (A. Wald) stepwise logistic regression analyses were used to determine the relationship between repolarization variables (covariates) and total or cardiovascular mortality (dependent variables). All data were evaluated by use of the program SPSS-PC+ (SPSS-PC+ Inc., Chicago, IL, USA) and *p* values of less than or equal to 0.05 were considered statistically significant.

## 3. Results

Starting from 255 eligible patients with symptoms of decompensated CHF, 12 patients were excluded due to poor-quality ECGs. Consequently, 243 CHF patients were included in the first part of the study. During the hospitalization, a total of 46 patients died (overall mortality rate, 19%): 23 (9.4%) died of bronchopneumonia and respiratory failure (1 patient from COVID-19 pneumonia), 17 died due to terminal heart failure (7%), 3 died of fatal myocardial infarction (1%), 3 died of sudden cardiac death (1%) (2 had sustained ventricular tachycardia and ventricular fibrillation; 1 had acute cor pulmonale secondary to massive embolism). The overall cardiovascular mortality rate was 9.4%. 

The general characteristics of HFrEF, HFmrEF, and HFpEF were significantly different based on the echocardiographic data (Table 1). 

In fact, the LVEF_Ech_ was significantly lower in HFrEF patients compared to the other two groups (*p* < 0.001) and was lower in HFmrEF than in HFpEF subjects (*p* < 0.001) (Table 1). Left ventricular mass index (*p* < 0.001), left ventricular end-diastolic diameter (*p* < 0.001), and left atrial transversal diameter (*p* < 0.05) were significantly higher in HFrEF in comparison with the other two patient groups (Table 1). Tricuspid annular plane systolic was lower in in HFrEF than HFpEF subjects (Table 1). Finally, tricuspid regurgitation peak gradient was higher in the HFrEF than HFpEF group (Table 1). HFrEF subjects reported a significantly higher blood level of NT-proBNP (*p* < 0.05) in comparison with HFpEF subjects (Table 1) while, on the contrary, the HFrHF group reported a significantly higher troponin level compared to both the other two groups (Table 1). The HFpEF subjects showed a significantly higher creatinine clearance in blood levels of than the other two groups (*p* < 0.05), but the HFmrEF group had a higher creatinine clearance than the HFrEF group (Table 1). Obviously, the HFrEF group included a significantly higher number of subjects with renal insufficiency than the HFpEF group (Table 1). A known history of ischemic heart disease was significantly predominant in subjects with HFrEF compared to the other two groups (HFmrEF: *p* < 0.05; HFpEF: *p* < 0.001) (Table 1). There were significantly more patients with left bundle branch block or with a pacemaker–ICD in the HFrEF and HFmrEF groups compared to the HFpEF group (*p* < 0.001) (Table 1). There were significantly more patients treated with beta-blockers and furosemide in the HFrEF category than in the HFpEF group. HFmrEF patients were also treated with beta-blockers to a greater extent than HFpHF patients (*p* < 0.05) (Table 1). Additionally, aldosterone antagonist use was significantly more common in patients with HFrEF than in those with HFpEF (*p* < 0.05), and the latter group received more dihydropyridine calcium channel blocker therapy than HfrEF patients (Table 1).

HFrEF patients reported a significant increase in QTe (*p* < 0.001), QTe_SD_ (*p* < 0.05), QTp (*p* < 0.05), Te (*p* < 0.001), and TeSD (*p* < 0.001) compared with HFpHF patients (Table 2). Only TeSD (*p* < 0.001) was significantly higher in HFmrEF (*p* < 0.05) than in HFpEF patients (Table 2). 

HR (*p* < 0.05) and EDFR (*p* < 0.05) were significantly higher in HFrEF than HfrEF patients (Table 3). On the contrary, SV (*p* < 0.05), SVI (*p* < 0.050), LVEF_BIO_ (*p* < 0.001), and ConI (*p* < 0.001) were significantly lower in subjects with HFpEF (Table 3). 

Deceased patients reported an increase in some repolarization data, especially the Te and Te_SD_ (Te: 120 ± 28 vs. 100 ± 25 ms, *p* < 0.001; Te_SD_: 9 vs. 7, *p*: 0.014). There was no statistical difference between the deceased or the survivors for the non-invasive hemodynamic data.

In the second study, the data were used to compare responders (n.: 60, mean age: 82 ± 7) and non-responders (n.: 30, 81 ± 11) to drug therapies, where the responders reported higher levels of NT-proBNP (4295 i.r. 7528 vs. 2110 i.r. pg/mL, *p* < 0.001) both at baseline and at the discharge (2030 i.r. 3250 vs. 5763 i.r. 7328 pg/mL, *p* < 0.001). In particular, the responders had a 46% reduction of NT-proBNP levels (from 4295 i.r. 7528 to 2030 i.r. 3250 pg/mL, *p* < 0.001). On the contrary, the non-responders had a 58% increase in NT-proBNP levels (from 2110 i.r. 5082 to 5763 i.r. 7328 pg/mL, *p* < 0.001). The repolarization data (Te and Te_SD_) were similar in both responders and non-responders at baseline (Te: 102 ± 30 vs. 99 ± 27 ms, *p*: ns; Te_SD_: 9 i.r. 5 vs. 8 i.r. 3 ms, *p*: ns) but were higher in the non-responder group at discharge (Te: 96 ± 23 vs. 108 ± 31 ms, *p* < 0.05; Te_SD_: 6 i.r. 3 vs. 10 i.r. 6 ms, *p* < 0.001) (Figure 2).

In the responder group, both Te and Te_SD_ decreased significantly (Te: from 102 ± 30 to 96 ± 23 ms, *p* < 0.05), and both of these values increased in the non-responder group (Te: from 99 ± 27 to 108 ± 31 ms, *p* < 0.05; Te_SD_: 8 i.r. 3 vs. 10 i.r. 6 ms, *p*: 0.001) (Figure 2). Regarding the non-invasive hemodynamic data, from baseline to discharge, the responder group had a significant decrease in heart rate (*p* < 0.05) SBP (*p* < 0.05), DBP (*p* < 0.05), and MBP (*p* < 0.05) (Table 4) and increased left ventricular ejection time (*p* < 0.05) (Table 4). The non-responder group did not show any difference in these two study conditions (Table 4). Finally, at the discharge, the responders reported higher levels LVEF_BIO_ (*p* < 0.05) and ConI (*p* < 0.05) than non-responders (Table 4), but the non-responders showed an increase in LVEDV (*p* < 0.05) (Table 4).

Multivariable logistic regression analysis reported that Te was related to an increase of total (*p* < 0.001) (Table 5) or cardiovascular mortality (*p* < 0.001) (Table 6).

## 4. Discussion

The major finding of the present study was confirmation of the association of the Te interval with an increased risk of in-hospital mortality in decompensated patients [17,18,19,20,22,23,43]. Secondly, a significant difference was not observed in ECG repolarization and non-invasive hemodynamic data between HFrEF and HFmrEF patients (Table 2 and Table 3). However, there was an increase in repolarization data only in HFrEF patients compared to the HFpEF group and only Te_SD_ was higher in the HFmrEF group in comparison with the HFpEF group (Table 2). The non-invasive hemodynamic data showed an obvious significant reduction of systolic function data (HR, SV, SVI, CI, LVEF_BIO_, and ConI) in patients with HFrEF in comparison to the HFpEF group (Table 3). For diastolic function data, we reported a marked reduction in EDFR in HFpEF in comparison with HFrEF patients (Table 3). Thirdly, only the responder patients showed a significant improvement of repolarization and specific hemodynamic data between baseline and discharge. Specifically, the data indicated a decrease in the Te, Te_SD_, HR, SBP, DBP, and MBP and an increase in the LVET (Figure 2) (Table 4). On the contrary, no variations were observed, between baseline and discharge, in non-responder groups for non-invasive hemodynamic data, but regarding repolarization data, there was an increase in Te and Te_SD_ (Figure 2). Finally, at discharge, responder patients demonstrated higher levels of LVEF_BIO_ and ConI and lower levels of LVEDV, Te, and Te_SD_. From this set of data, it can be seen that Te remains a useful short-term marker of hospital mortality, but repolarization and non-invasive hemodynamic data were not able to identify the specific categories of CHF based on LVEF_ECO_. However, the low level of agreement between LVEF_ECH_ and LVEF_BIO_ has been previously reported by our group in the same study patients [19]. Conversely, repolarization data and non-invasive hemodynamic data could be used to monitor patients with CHF because this approach allows the use of clinical signs to identify non-responders. In particular, the lack of change in Te, Te_SD_, heart rate, blood pressure, and LVET, which are a heart-rate-dependent parameters [44], were the best markers of NT-proBNP reduction (Table 4). However, the best results were obtained with repolarization data. The Te and Te_SD_ data showed divergent behavior among non-responders. At the time of discharge, the responders had a significant reduction, while the non-responders had increases in these parameters (Figure 2).

In this study, we discovered the Te and Te_SD_ as specific markers of mortality and CHF decompensation, but the pathophysiological basis of these ECG parameters is still controversial. In fact, in the past, this last part of repolarization was extensively studied as a marker of sudden arrhythmic death both in vivo and in vitro. In particular, some authors hypothesized that the Te interval could represent the transmural dispersion of repolarization and this electrophysiologic condition was related to ventricular malignant arrhythmias [45,46]. This hypothesis has been scrutinized but it is still erroneously reported in studies as the probable electrophysiological basis of the Te [47,48,49]. However, this hypothesis does not explain how the Te interval is widely associated with an increased risk of total and non-arrhythmic mortality [14,15,16]. Our data indicate the Te interval was an independent risk factor for total mortality, cardiovascular mortality, and CHF decompensation in a study population with a low level of sudden arrhythmic death. We believe sympathetic hyperactivity and neurohumoral activation could influence the Te and TeSD. In previous studies, it was reported that sympathetic activation during exercise was able to induce an increase in Te and TeSD in normal subjects [50]. In animal models, we observed an increase in the short-term variability of the Te interval during pacing-induced acute congestive heart failure and experimental acute myocardial ischemia [35,51]. Additionally, the temporal dispersion markers of Te were strongly correlated to the stellate ganglion nerve activity [35,51]. Subsequently, another study reported an increase in the Te interval during left, right, and bilateral stellate ganglion stimulations [52]. Therefore, experimental and clinical studies individuated a relation between sympathetic activity and Te interval. We believe that sympathetic hyperactivity in CHF could affect the last part of repolarization by altering voltage-gated ion channels. We hypothesized that ion channel remodeling in CHF could be capable of increasing both duration and temporal dispersion, expressed as standard deviations of the Te interval. In the last thirty years, experimental and electrophysiological studies have established the deep involvement of sodium channels, potassium channels, calcium handling, and currents in CHF resulting in an increase in the action potential duration [53]. In particular, a delayed inactivation of sodium channels with a prolonged inward of sodium beyond phase 0 was reported [54,55]. Contrarily, a down-regulation of the potassium channels (Ito, IKs, IKr, IK1, and IK2P) [54] represents the most important outward flow of positive ions capable of regulating the action potential duration. Finally, a considerable number of studies indicated impairing of the intracellular myocardial calcium cycle by a multilevel involvement of structures, enzymes, and ion channels resulting in a cytosolic calcium overload, a reduced calcium sarcoplasmic reticulum content, and an increase in action potential duration [56]. On the other hand, in addition, the pathologic myocardial substrate could play a leading role. In fact, myocardial ischemia, necrosis, fibrosis, hypertrophy, and fiber disarray could undoubtedly induce an increase in action potential duration and more specifically Te and Te_SD_. 

The use of artificial intelligence (AI) in the field of clinical cardiology is increasingly promising [57,58,59,60]. In particular, in heart failure, the use of knowledge in the pathophysiological and electrocardiographic fields, combined with the possibility of remote monitoring, can play a fundamental role in the lives of patients suffering from this complex clinical condition [61,62,63,64,65,66]. Thus, machine learning tools are extremely important to acquire deep knowledge and reach specific stratification–prognostic models. The possibility of intercepting clinical alterations in advance, predicted by the electrocardiographic changes of patients suffering from CHF, would allow the cardiologist to modify therapy in time and, hopefully, reduce the possibility of acute heart failure.

Furthermore, the use of machine learning would be of enormous importance in being able to refine the diagnostic capabilities of AI tools with the data collected over time [67,68]. In conclusion, these simple, inexpensive, non-invasive, easily repeatable, and transmissible markers coupled with autonomous decision-making processes based on artificial intelligence algorithms could improve the management and the prognosis of these frail patients.

## 5. Limitations

The present study is burdened by the smallness of the sample evaluated, albeit calculated a priori. 

The small sample size and the advanced age of the enrolled patients influenced the possibility of analyzing the data obtained by correlating them to the dosage of drugs, usually taken by patients for chronic heart failure therapy. In fact, basically faithful to the geriatric medicine principle of “start low and go slow”, almost all patients were taking low doses of drugs. Larger studies could allow observation of differences and stratify patients according to drug dosage.

A further limitation was studying hospitalized patients and hypothesizing the use of the data obtained for patients not yet hospitalized. Further studies on outpatient populations are necessary.

Furthermore, another limitation of the study is that just one patient was treated with SGLT2 inhibitors. The sample was in fact largely studied before the recent indications provided by the European Society of Cardiology guidelines on the use in class I evidence A of these drugs in subjects with heart failure and diabetes mellitus [7]. In fact, with the use of these drugs, we expect a lower frequency of hospitalizations for acute cardiac decompensation in CHF patients and, having a fundamentally diuretic effect, even a lower retention of liquids recognizable to bioimpedance. Further enrollment will help fill this gap.

## Figures and Tables

**Figure 1 biomedicines-12-00716-f001:**
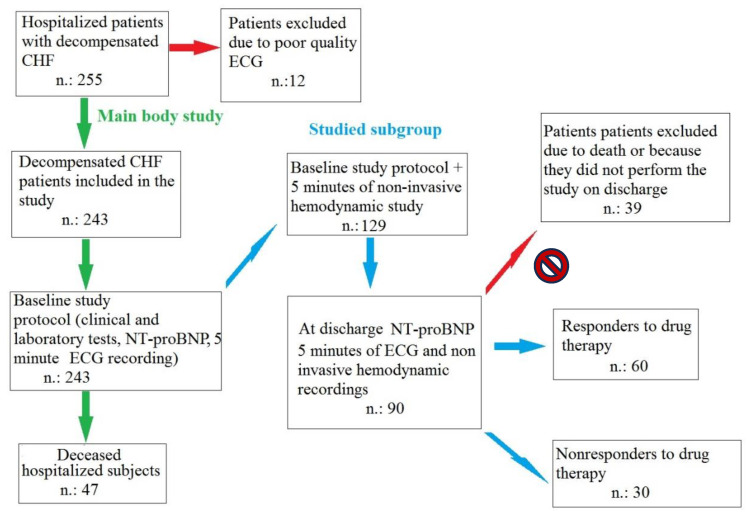
Flowchart of study phases.

**Figure 2 biomedicines-12-00716-f002:**
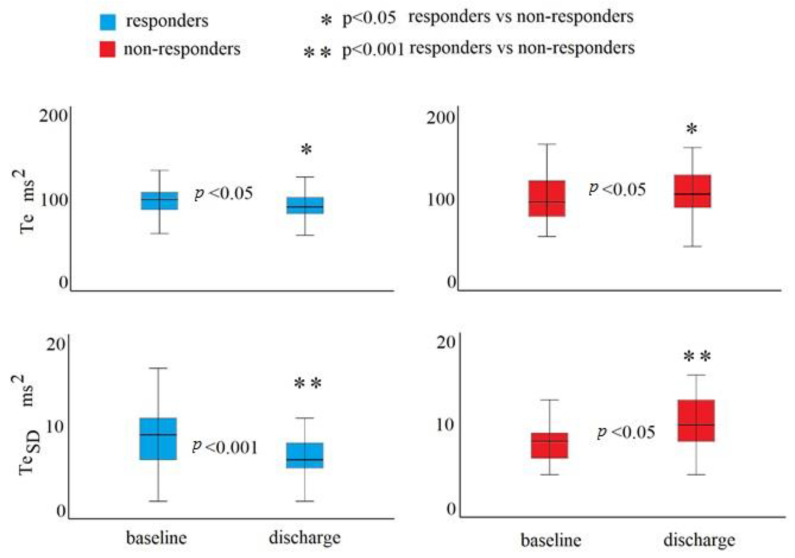
Tpeak−Tend intervals’ mean (Te) and standard deviation (TeSD) of a 5 min ECG recording at baseline and discharge.

**Table 1 biomedicines-12-00716-t001:** General characteristics of the studied population.

	Heart Failure with Reduced Ejection Fraction	Heart Failure with Mildly Reduced Ejection Fraction	Heart Failure with Preserved Ejection Fraction	
	N: 112	N: 38	N: 93	*p*
Gender, M/F	65/47	16/22	41/52	>0.5
Age, years	83 ± 10	85 ± 7	83 ± 10	0.497
BMI, kg/m^2^	26 ± 5	25 ± 4	26 ± 5	0.536
Heart Rate (radial pulse), beats/m	74 ± 12	73 ± 15	76 ± 13	0.463
Systolic Blood Pressure, mmHg	120 ± 17 **	127 ± 17	130 ± 21 **	<0.001
Diastolic Blood Pressure, mmHg	67 ± 9 **	67 ± 11	73 ± 11 **	<0.001
Left Ventricular Ejection Fraction, %	32 ± 7 **##	45 ± 1 ##§§	52 ± 3 **§§	<0.001
Left Ventricular Mass Index, g/m^2^	147 ± 34 **#	129 ± 34 #	123 ± 26 **	<0.001
Left Ventricular End-Diastolic Diameter, mm	57 ± 8 **##	51 ± 5 ##	50 ± 5 **	<0.001
Posterior Wall Thickness, mm	11 ± 2	11 ± 2	11 ± 2	0.691
Interventricular Septum Thickness, mm	12 ± 2	12 ± 1	12 ± 1	0.594
Left Atrial Transversal Diameter, mm	49 ± 8 *#	45 ± 6 #	45 ± 6 *	<0.001
Tricuspid Annular Plane Systolic Excursion, mm	19 ± 3 **	21 ± 5	22 ± 4 **	<0.001
Tricuspid Regurgitation Peak Gradient, mmHg	47 ± 13 #	40 ± 11 #	43 ± 15	0.030
Hemoglobin (g/dL)	11.4 ± 2.1	11.1 ± 1.6	11.4 ± 2.1	0.689
Arterial O_2_ Saturation, %	97 ± 3	97 ± 2	97 ± 4	0.655
Fraction of Inspired O_2_,%	27 ± 9	26 ± 7	26 ± 8	0.828
PaO_2_/FiO_2_ ratio	334 ± 103	339 ± 126	329 ± 90	0.884
A-ADO_2_, mmHg	39 [51]	35 [49]	33 [41]	0.671
NT-proBNP, pg/mL	6170 [9995] *	2725 [4590]	1660 [3085] *	<0.001
C-ReactiveProtein (mg/dL)	4.0 [9.4]	3.6 [11.7]	5.1 [8.8]	0.926
High-Sensitivity Cardiac Troponin/(pg/L)	62 [98] **##	36 [47] ##	31 [29] **	<0.001
Serum Sodium (mmol/L)	141 ± 6	141 ± 5	141 ± 5	0.957
Serum Potassium (mmol/L)	4.0 ± 0.6	4.1 ± 0.6	4.1 ± 0.6	0.532
Serum Calcium (mmol/L)	2.1 ± 0.2	2.2 ± 0.2	2.2 ± 0.2	0.054
Creatinine Clearance (mL/m)	38 [24] *#	48 [30] #§	50 [38] *§	0.025
Serum Urea (mmol/L)	12.3 [8.3]	10.2 [7.8]	8.3 [5.6]	0.003
Albumin (g/dL)	3.4 ± 0.6	3.4 ± 0.5	3.5 ± 0.6	0.736
Fasting Glucose (mmol/L)	6.8 ± 2.2	6.6 ± 2.7	6.2 ± 2.7	1.131
HbA_1c_ (%)	6.2 ± 1.2	5.9 ± 1.3	5.8 ± 1.0	0.099
Total Cholesterol (mmol/L)	3.6 ± 0.9	3.8 ± 0.9	3.9 ± 1.0	0.444
HDL-Cholesterol (mmol/L)	1.1 ± 0.4	1.1 ± 0.4	1.1 ± 0.4	0.975
LDL-Cholesterol (mmol/L)	1.9 ± 0.8	2.1 ± 0.6	2.1 ± 0.9	0.608
Triglycerides (mmol/L)	1.9 ± 1.6	1.6 ± 0.7	1.5 ± 0.7	0.219
Hypertension, n (%)	80 (71)	32 (84)	77 (41)	0.087
Hypercholesterolemia, n (%)	54 (48)	14 (37)	41 (44)	0.468
Diabetes, n (%)	57 (51)	12 (32)	32 (34)	0.087
Renal Insufficiency, n (%)	67 (60) *	16 (42)	37 (40) *	0.010
Known Myocardial Ischemia History, n (%)	57 (51) **#	8 (21) #	18 (19) **	<0.001
Valve Diseases, n (%)	26 (23)	12 (32)	24 (25)	0.651
Premature Supraventricular Complexes, n (%)	12 (11)	2 (5)	9 (10)	0.609
Premature Ventricular Complexes, n (%)	26 (23)	11 (29)	16 (17)	0.298
Permanent Atrial Fibrillation, n (%)	46 (41)	18 (47)	30 (32)	0.213
Left Bundle Branch Block, n (%)	34 (30) **	10 (26) §§	6 (7) **§§	<0.001
Right Bundle Branch Block, n (%)	18 (16)	4 (11)	13 (14)	0.694
Pacemaker–ICD, n (%)	37 (33) **	10 (26) §§	8 (9) **§§	<0.001
Deceased Hospitalized Patients, n (%)	27 (24)	7 (18)	13 (14)	0.186
β-Blockers, n (%)	82 (73) *	30 (80) §	52 (56) *§	0.008
Furosemide, n (%)	97 (87) **	31 (82)	60 (65) **	0.001
ACE/Sartans	45 (40)	10 (26)	41 (44)	0.165
Aldosterone Antagonists, n (%)	25 (22) *	7 (18)	8 (9) *	0.029
Potassium, n (%)	7 (6)	2 (5)	8 (9)	0.726
Nitrates, n (%)	14 (13)	6 (16)	8 (9)	0.458
Digoxin, n (%)	6 (5)	3 (8)	3 (3)	0.514
Statins, n (%)	36 (32)	8 (21)	26 (28)	0.416
Antiplatelet Drugs, n (%)	47 (42)	9 (24)	35 (38)	0.132
Oral Anticoagulants, n (%)	27 (24)	13 (34)	30 (32)	0.319
Diltiazem or Verapamil, n (%)	1 (1)	1 (3)	6 (7)	0.082
Ivabradine, n	2 (2)	1 (3)	2 (2)	0.948
Dihydropyridine Calcium Channel Blockers, n (%)	8 (7) *	6 (16)	17 (18) *	0.049
Propafenone, n (%)	0 (0)	0 (0)	2 (2)	0.197
Amiodarone, n (%)	11 (10)	1 (3)	7 (8)	0.358
Valsartan/Sacubitril, n (%)	4 (4)	0 (0)	0 (0)	0.093
SGLT-2i, n (%)	1 (1)	0 (0)	0 (0)	0.999

Data are expressed as mean ± SD, or median [interquartile range], or number of patients (%); ** *p* < 0.001 Heart Failure with Reduced Ejection Fraction versus Heart Failure with Preserved Ejection Fraction; * *p* < 0.05 Heart Failure with Reduced Ejection Fraction versus Heart Failure with Preserved Ejection Fraction; ## *p* < 0.001 Heart Failure with Reduced Ejection Fraction versus Heart Failure with Mildly Reduced Ejection Fraction; # *p* < 0.05 Heart Failure with Reduced Ejection Fraction versus Heart Failure with Mildly Reduced Ejection Fraction; §§ *p* < 0.001 Heart Failure with Mildly Reduced Ejection Fraction versus Heart Failure with Preserved Ejection Fraction; § *p* < 0.05 Heart Failure with Mildly Reduced Ejection Fraction versus Heart Failure with Preserved Ejection Fraction.

**Table 2 biomedicines-12-00716-t002:** ECG Data of Study Subjects.

	Heart Failure with Reduced Ejection Fraction	Heart Failure with Mildly Reduced Ejection Fraction	Heart Failure with Preserved Ejection Fraction	
Variables	N: 112	N: 38	N: 93	*p*
RR, ms	850 ± 160	864 ± 164	871 ± 174	0.671
QTe, ms	490 ± 87 **	457 ± 94	427 ± 65 **	<0.001
QTe_SD_, ms	10 [5] *	10 [5]	8 [5] *	0.043
QTp, ms	373 ± 85 *	353 ± 83	332 ± 56 *	0.001
QTp_SD_, ms	9 [5]	9 [4]	8 [3]	0.216
Te, ms	110 ± 27 **	106 ± 31	94 ± 22 **	<0.001
Te_SD_, ms	8 [6] **	9 [6] §	6 [4] **§	<0.001

Data are expressed as mean ± SD, or median [interquartile range]; ** *p* < 0.001 Heart Failure with Reduced Ejection Fraction versus Heart Failure with Preserved Ejection Fraction; * *p* < 0.05 Heart Failure with Reduced Ejection Fraction versus Heart Failure with Preserved Ejection Fraction; § *p* < 0.05 Heart Failure with Mildly Reduced Ejection Fraction versus Heart Failure with Preserved Ejection Fraction.

**Table 3 biomedicines-12-00716-t003:** Non-invasive Hemodynamic Data of Study Subjects.

	Heart Failure with Reduced Ejection Fraction	Heart Failure with Mildly Reduced Ejection Fraction	Heart Failure with Preserved Ejection Fraction	
Variables	N: 63	N: 20	N: 46	*p*
Heart Rate, b/m	81 ± 23 *	86 ± 30 §	72 ± 13	0.031
Stroke Volume, mL	59 ± 17 *	64 ± 23	71 ± 20 *	0.008
Stroke Volume Index, mL/m^2^	33 ± 10 *	37 ± 14	40 ± 10 *	0.004
Cardiac Output, L/m	4.59 ± 1.39	5.04 ± 1.60	4.97 ± 1.16	0.239
Cardiac Index, L/m/m^2^	2.53 ± 0.75 *	2.95 ± 0.87	2.81 ± 0.74 *	0.047
Systemic Vascular Resistance, Dyn.s/cm^2^	3390 ± 1440	2901 ± 739	2932 ± 877	0.090
Systemic Vascular Resistance Index, Dyn.s/cm^2^.m^2^	1849 ± 692	1716 ± 458	1639 ± 491	0.188
SBP, mmHg	122 ± 17	124 ± 13	125 ± 13	0.503
MBP, mmHg	71 ± 10	73 ± 9	72 ± 10	0.691
DBP, mmHg	93 ± 12	95 ± 9	95 ± 10	0.563
Left Ventricular Ejection Fraction, %	34 ± 13 **	39 ± 14	46 ± 15 **	<0.001
Contractility Index	61 ± 40 *	78 ± 44	86 ± 53 *	0.031
Left Ventricular Ejection Time, ms	267 ± 77	271 ± 74	291 ± 86	0.271
Left Cardiac Work Index, kg.m/m^2^	3.04±	3.63 ± 1.20	3.49 ± 1.04	0.063
Left Ventricular End-Diastolic Volume, mL	194 ± 90	195 ± 134	164 ± 49	0.162
Early Diastolic Filling Ratio	92 ± 35 *	98 ± 57	77 ± 25 *	0.045

Data are expressed as mean ± SD, or median [interquartile range]; ** *p* < 0.001 Heart Failure with Reduced Ejection Fraction versus Heart Failure with Preserved Ejection Fraction; * *p* < 0.05 Heart Failure with Reduced Ejection Fraction versus Heart Failure with Preserved Ejection Fraction; § *p* < 0.05 Heart Failure with Mildly Reduced Ejection Fraction versus Heart Failure with Preserved Ejection Fraction.

**Table 4 biomedicines-12-00716-t004:** Clinical, ECG, and Non-invasive Hemodynamic Data of Study Subjects at Baseline and at the Discharge.

	Responders			Non-Responders		
	Baseline	Discharge		Baseline	Discharge	
Variables	N: 60	N: 60	*p*	N: 30	N: 30	*p*
Heart Rate, b/m	82 ± 26	74 ± 17	0.020	74 ± 17	77 ± 19	0.308
Stroke Volume, mL	64 ± 20	66 ± 24	0.490	65 ± 15	63 ± 15	0.576
Stroke Volume Index, mL/m^2^	36 ± 12	38 ± 14	0.536	35 ± 9	35 ± 10	0.445
Cardiac Output, L/m	4.88 ± 1.48	4.72 ± 1.73	0.558	4.58 ± 1.09	4.60 ± 1.24	0.974
Cardiac Index, L/m/m^2^	2.79 ± 0.87	2.69 ± 1.02	0.520	2.53 ± 0.63	2.52 ± 0.62	0.840
Systemic Vascular Resistance, Dyn.s/cm^2^	3099 ± 1279	3049 ± 1113	0.868	3221 ± 1048	3307 ± 1166	0.747
Systemic Vascular Resistance Index, Dyn.s/cm^2^.m^2^	1748 ± 600	1730 ± 619	0.795	1760 ± 506	1852 ± 783	0.532
SBP, mmHg	124 ± 14	119 ± 13	0.025	120 ± 15	123 ± 14	0.190
MBP, mmHg	94 ± 9	90 ± 10	0.006	92 ± 11	94 ± 14	0.297
DBP, mmHg	71 ± 10	68 ± 10	0.016	71 ± 10	72 ± 9	0.693
Left Ventricular Ejection Fraction, %	40 ± 16	41 ± 17 *	0.688	36 ± 14	34 ± 14 *	0.427
Contractility Index	78 ± 56	86 ± 56 *	0.296	67 ± 36	58 ± 29 *	0.108
Left Ventricular Ejection Time, ms	271 ± 98	299 ± 91	0.019	275 ± 60	264 ± 80	0.631
Left Cardiac Work Index, kg.m/m^2^	3.39 ± 1.29	3.14 ± 1.44	0.300	3.05 ± 0.97	3.06 ± 0.89	0.459
Left Ventricular End-Diastolic Volume, mL	172 ± 77	171 ± 64 *	0.907	196 ± 71	200 ± 73 *	0.459
Early Diastolic Filling Ratio	91 ± 45	81 ± 25	0.520	88 ± 32	91 ± 47	0.471

Data are expressed as mean ± SD, or median [interquartile range]; * *p* < 0.05 responders versus non-responders.

**Table 5 biomedicines-12-00716-t005:** Logistic Regression Between in-Hospital Total Mortality (dependent variable) and ECG Repolarization Data.

Variables	χ^2^	B	Univariable Analysis Odds Ratio (95% CI)	*p* Values	χ^2^	B	Multivariable Analysis Odds Ratio (95% CI)	*p* Values
					35.45			
QTe	0.009	0.00	1.00 (1.00–1.00)	0.924		−0.003	1.00 (0.98–1.01)	0.429
QTe_SD_	2.90	0.04	1.04 (0.99–1.09)	0.096		0.057	1.06 (0.98–1.14)	0.142
QTp	2.82	−0.01	1.00 (0.99–1.00)	0.058		−0.005	1.00 (0.99–1.00)	0.206
QTp_SD_	0.36	0.02	1.02 (0.95–1.11)	0.542		−0.086	0.92 (0.81–1.05)	0.198
Te	19.49	0.03	1.03 (1.01–1.04)	<0.001		0.032	1.03 (1.01–1.05)	0.001
Te_SD_	7.92	0.07	1.07 (1.01–1.14)	0.027		0.047	1.05 (0.98–1.12)	0.141

**Table 6 biomedicines-12-00716-t006:** Logistic Regression Between in-Hospital Cardiovascular Mortality (dependent variable) and ECG Repolarization Data.

Variables	χ^2^	B	Univariable Analysis Odds Ratio (95% CI)	*p* Values	χ^2^	B	Multivariable Analysis Odds Ratio (95% CI)	*p* Values
					32.58			
QTe	0.54	0.00	1.00 (1.00–1.01)	0.454		−0.01	1.00 (0.99–1.01)	0.453
QTe_SD_	1.97	0.05	1.05 (0.98–1.13)	0.145		0.06	1.07 (0.94–1.21)	0.317
QTp	2.10	−0.00	1.00 (0.99–1.00)	0.153		−0.01	1.00 (0.99–1.00)	0.187
QTp_SD_	0.67	0.04	1.04 (0.95–1.14)	0.401		−0.09	0.92 (0.77–1.09)	0.317
Te	20.83	0.03	1.03 (1.02–1.05)	<0.001		0.036	1.04 (1.02–1.06)	0.001
Te_SD_	8.77	0.08	1.08 (1.01–1.16)	0.024		0.052	1.05 (1.98–1.03)	0.160

## Data Availability

All data, materials, and codes used in this study are available upon request from the corresponding author.

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
