# Peer review of "Artificial Intelligence Applied to Electrical and Non-Invasive Hemodynamic Markers in Elderly Decompensated Chronic Heart Failure Patients"

_biomedicines, 2024, doi:10.3390/biomedicines12040716_

Round 1

Reviewer 1 Report

Comments and Suggestions for Authors

I have carefully reviewed the manuscript titled “Artificial Intelligence Applied to Electrical and Noninvasive Hemodynamic Markers in Decompensated Chronic Heart Failure Patients with or without Reduced Ejection Fraction” and would like to provide my feedback. Overall, the study presents an interesting investigation into the predictive power of Tend interval and non-invasive hemodynamic markers in decompensated chronic heart failure (CHF) patients. However, the manuscript would benefit from further clarification and revision to ensure its rigor and repeatability.

 1.     The study objectives are clearly stated, with the aims of the study to (1) assess the predictive power of Tend interval (Te) and non-invasive hemodynamic markers based on bioimpedance in decompensated CHF patients; (2) to investigate the differences in repolarization and hemodynamic data among CHF patients grouped based on different levels of left ventricular ejection fraction (LVEF); (3) To determine if repolarization and hemodynamic data change with clinical improvement or worsening in CHF patients. However, the objectives and rationale of the study are mixed and need further clarification to highlight the strengths of their study.

2.     In terms of methods, the manuscript states that 243 decompensated CHF patients were studied using 5-minute ECG recordings to determine the mean and standard deviation of Te in the first study. In the second study, a subgroup of 129 patients had non-invasive hemodynamic and repolarization data recorded for further evaluation. However, several methodological details need to be clarified for better understanding and reproducibility.

-        For example, details on how to avoid selection bias, ethical approval number, data collection procedures, and statistical analyses need to be provided in more depth. This will ensure that other researchers can replicate the study and validate the findings.

-        The details for how the non-invasive hemodynamic parameters were obtained and evidence of their accuracy (for example by providing adequate references), since this is a major novelty of the current study.

 3.     The results section presents important findings related to in-hospital and cardiovascular mortality rates, with Te being higher in deceased subjects compared to surviving subjects. The multivariable logistic regression analysis demonstrates a significant relationship between Te and both total and cardiovascular mortality. Additionally, the results show that subjects with heart failure with reduced ejection fraction (HFrEF) exhibit higher levels of repolarization and lower non-invasive systolic hemodynamic data compared to those with preserved ejection fraction (HFpEF). Furthermore, patients in the subgroup with NT-proBNP reduction after therapy demonstrated lower levels of Te, heart rate, blood pressure, contractility index, and left ventricular ejection time compared to patients without NT-proBNP reduction.

4.     In conclusion, the manuscript presents valuable insights into the use of electrical signals from ECG and bioimpedance as monitoring tools for patients with decompensated CHF. The non-invasive and easily repeatable markers discussed in the study have the potential to facilitate remote monitoring and early intervention for the possible worsening of these patients using machine learning and artificial intelligence tools.

5.     The limitation is not sufficient to discuss potential sources of bias in data collection, data for the use of diuretics, and the lack of invasive hemodynamic comparison of the study, acknowledging these limitations would provide a balanced perspective on the implications of their findings.

Author Response

#Reviewer 1

We would like to thank the reviewer for his/her time spent for the revision. We have done our best to fulfill the request.

I have carefully reviewed the manuscript titled “Artificial Intelligence Applied to Electrical and Noninvasive Hemodynamic Markers in Decompensated Chronic Heart Failure Patients with or without Reduced Ejection Fraction” and would like to provide my feedback. Overall, the study presents an interesting investigation into the predictive power of Tend interval and non-invasive hemodynamic markers in decompensated chronic heart failure (CHF) patients. However, the manuscript would benefit from further clarification and revision to ensure its rigor and repeatability.

  1. The study objectives are clearly stated, with the aims of the study to (1) assess the predictive power of Tend interval (Te) and non-invasive hemodynamic markers based on bioimpedance in decompensated CHF patients; (2) to investigate the differences in repolarization and hemodynamic data among CHF patients grouped based on different levels of left ventricular ejection fraction (LVEF); (3) To determine if repolarization and hemodynamic data change with clinical improvement or worsening in CHF patients. However, the objectives and rationale of the study are mixed and need further clarification to highlight the strengths of their study.

Reply: Thank you for your suggestions, we have tried to better clarify our scopes.

  1. In terms of methods, the manuscript states that 243 decompensated CHF patients were studied using 5-minute ECG recordings to determine the mean and standard deviation of Te in the first study. In the second study, a subgroup of 129 patients had non-invasive hemodynamic and repolarization data recorded for further evaluation. However, several methodological details need to be clarified for better understanding and reproducibility.

-        For example, details on how to avoid selection bias, ethical approval number, data collection procedures, and statistical analyses need to be provided in more depth. This will ensure that other researchers can replicate the study and validate the findings.

Reply: Thank you for your precious suggestions. We have added two sentences in order to provide further information.

-        The details for how the non-invasive hemodynamic parameters were obtained and evidence of their accuracy (for example by providing adequate references), since this is a major novelty of the current study.

Reply: Thanks for highlighting this point. We published a work 2 years ago in Biomedicines on the use of bioimpedance measurement in risk stratification of adverse events during the exacerbation of acute heart failure. As well as, in this latest work we have presented in the bibliography some of the papers that inspired our research (references 23-26). We have inserted quote 19 in the methods paragraph so as to refer to our previous work.

  1. The results section presents important findings related to in-hospital and cardiovascular mortality rates, with Te being higher in deceased subjects compared to surviving subjects. The multivariable logistic regression analysis demonstrates a significant relationship between Te and both total and cardiovascular mortality. Additionally, the results show that subjects with heart failure with reduced ejection fraction (HFrEF) exhibit higher levels of repolarization and lower non-invasive systolic hemodynamic data compared to those with preserved ejection fraction (HFpEF). Furthermore, patients in the subgroup with NT-proBNP reduction after therapy demonstrated lower levels of Te, heart rate, blood pressure, contractility index, and left ventricular ejection time compared to patients without NT-proBNP reduction.

Reply: You have fully understood the points! Thank you.

  1. In conclusion, the manuscript presents valuable insights into the use of electrical signals from ECG and bioimpedance as monitoring tools for patients with decompensated CHF. The non-invasive and easily repeatable markers discussed in the study have the potential to facilitate remote monitoring and early intervention for the possible worsening of these patients using machine learning and artificial intelligence tools.

Reply: these are the major findings. The noninvasive tools could allow us in the near future to intercept heart failure decompensations before the need to access emergency departments.

  1. The limitation is not sufficient to discuss potential sources of bias in data collection, data for the use of diuretics, and the lack of invasive hemodynamic comparison of the study, acknowledging these limitations would provide a balanced perspective on the implications of their findings.

Reply: Thank you! We feel to have adequately afforded the limitations of the study. The enrolled population was composed by elderly subjects who have specific characteristics. To better balance these aspects, we have modified the title.

Reviewer 2 Report

Comments and Suggestions for Authors

The  topic  of  the  study is  of  great importance. Notable, opportunities of single lead ECG analysis in heart failure give a perspective for noninvasive  monitoring in these patients. However, the paper requires several corrections.

1)      The core question is the aim of the study. From my point of view, several different studies are included in one  paper. Namely,

1.       Differences in   ECG data between HFREF, HFPEF and  HFmrEF

2.       Influence  of  basal ECG on  prognosis in hospitalized  patients  with  ADHF

3.       Correlation  of  ECG data  and NT-proBNP   between admission and  discharge in  patients  with  ADHF

The  aim of the  study have  to be  corrected to become  more clear. For  example, analyze  ECG  and  hospital prognosis in  ADHF.  ECG data  between  HF  subtypes could  be  published  separately or  just mentioned with  tables  in supplementary materials.  Otherwise, understanding  for  readers  is questionable

2)      Figure 1 requires clarification in numbers (30 +30+  39  vs  60)

3)      It is not correctly to name the patients who did not decrease NT-proBNP to the end of hospitalization as non-responders. 5-10 days of hospitalization is not enough for such conclusion. Better to use other term

4)      The name  of the paper is  about  AI. Thus some correction is required  in Methods section. Otherwise the  name have  to be  corrected.

5)      Table 1  is  abundant.  I recommend to limit up to 7-20 paraments and  present  other data in supplementary

6)      Table  3  is  about hemodymanic data. Not  ECG. Could be  presented in supplementation

7)      Outcome section is  required that should include length of stay, ICU admission as  mortality

8)      Lines 192-196  and 225-227  are about mortality. At  first, I recommend to start  with  mortality and  only in the  end  the  results  section analyze  NT-proBNP dynamic.  Otherwise finish with  mortality

Second. Cox analysis should  be  performed  for  mortality.

Third. Analysis could  be adjusted even for  age  and  sex.  Better-  for  EF etc

Fourth. Patients  were  divided  by EF  into HFREF, HFPEF and  HFmrEF.  Could  you  analyses  mortality rate  and  influence of Te and TeSD on it in these  subgroups

9)      NT-proBNP dynamics represents possible correlation or trend correlation between  NT-proBNP and Te. This could a  very important conclusion for  future studies for  distant  monitoring  with   single-lead  ECG derived Te and  Tesd  as a markers  of  hemodynamic congestion.  Thus  if  you leave  this  part  in  the  present  paper , I recommend  to correlate  NT-proBNP with    Te etc.   Moreover, this  correlation could  be  supported in each subgroup (HFREF, HFPEF and  HFmrEF).

10)   Discussion section should  be  corrected accorded to the  Results  section.

11)    Presented data was derived from  in-hospital course and  couldn’t be  translated into outpatients. Should  be  mentioned  in  limitations

12)     Abstract  should  be  corrected  after changes  in the  main body of  the  paper

Comments on the Quality of English Language

Proofreading  is  required

Author Response

#Reviewer 2

We would like to thank the reviewer for his/her time spent for the revision. We have done our best to fulfill the request.

The  topic  of  the  study is  of  great importance. Notable, opportunities of single lead ECG analysis in heart failure give a perspective for noninvasive  monitoring in these patients. However, the paper requires several corrections.

1)      The core question is the aim of the study. From my point of view, several different studies are included in one  paper. Namely,

  1. Differences in   ECG data between HFREF, HFPEF and  HFmrEF
  2. Influence  of  basal ECG on  prognosis in hospitalized  patients  with  ADHF
  3. Correlation  of  ECG data  and NT-proBNP   between admission and  discharge in  patients  with  ADHF

 Reply: Thank you for this suggestion. We have already published study 2 and study 3 you have hypothesized, so in this one we tried to go deeper and analyzed haemodynamic, ECG and biohumoral variables all together.

The  aim of the  study have  to be  corrected to become  more clear. For  example, analyze  ECG  and  hospital prognosis in  ADHF.  ECG data  between  HF  subtypes could  be  published  separately or  just mentioned with  tables  in supplementary materials.  Otherwise, understanding  for  readers  is questionable.

Reply: we have modified the title according to reviewer 1 suggestions.

2)      Figure 1 requires clarification in numbers (30 +30+  39  vs  60)

Reply: following the arrows’ colors, the red one was that of excluded patients, the blue-sky arrows conduct to 60 and 30 patients enrolled (tot. 90). Nevertheless, we have added a specific sign in order to clarify.

3)      It is not correctly to name the patients who did not decrease NT-proBNP to the end of hospitalization as non-responders. 5-10 days of hospitalization is not enough for such conclusion. Better to use other term.

Reply: Thank you for this observation. Our patients are elderly and frail, frequently and unfortunately the inhospital stay is longer than 10 days, not only for medical conditions/compications, but also for social problems which require to be solved before the discharge.

4)      The name  of the paper is  about  AI. Thus, some correction is required  in Methods section. Otherwise the  name have  to be  corrected.

Reply: We have added some sentences at the end of introduction and specifically “The final aim of this work is to identify the electrical, bioimpedance and biohumoral variables which, through artificial intelligence and machine learning devices, will be able to predict the clinical exacerbation of heart failure in the preliminary phases. By intercepting it early, it will be possible to treat it in the very early stages, trying to avoid hospitalization and the health and non-medical costs that would arise from it.”

5)      Table 1  is  abundant.  I recommend to limit up to 7-20 paraments and  present  other data in supplementary

Reply: You are right, we feel your point. Nevertheless, to be absolutely clear, we prefer to maintain these data in the main text, as well as those in table 3.

6)      Table  3  is  about hemodymanic data. Not  ECG. Could be  presented in supplementation

Reply: thank you! We have amended.

7)      Outcome section is  required that should include length of stay, ICU admission as  mortality.

Reply: these data have been already published Clinical cardiology, 45(12), 1192–1198. https://doi.org/10.1002/clc.23888.

8)      Lines 192-196  and 225-227  are about mortality. At  first, I recommend to start  with  mortality and  only in the  end  the  results  section analyze  NT-proBNP dynamic.  Otherwise finish with  mortality.

Reply: thank you, we understand what you mean, but this is not a study on “mortality” at all, but the aim was to identify the most impaired ECG, Haemodynamic variable to include in the near future in AI model, in order to intercept decompensating outpatients. So, mortality is in the background and has not the main role.

Second. Cox analysis should  be  performed  for  mortality.

Reply: se the previous point. We do not analyzed mortality in time, but just as an event. Mortality has been already addressed in a previous study J Clin Med. 2020 Jun 16;9(6):1879. doi: 10.3390/jcm9061879.

Third. Analysis could  be adjusted even for  age  and  sex.  Better-  for  EF etc

Reply: there were no differences for age and sex in the studied group. So, these variables have not the need to be separately analysed.

Fourth. Patients  were  divided  by EF  into HFREF, HFPEF and  HFmrEF.  Could  you  analyses  mortality rate  and  influence of Te and TeSD on it in these  subgroups

Reply: this could be a very interesting point to address in our next study!

9)      NT-proBNP dynamics represents possible correlation or trend correlation between  NT-proBNP and Te. This could a  very important conclusion for  future studies for  distant  monitoring  with   single-lead  ECG derived Te and  Tesd  as a markers  of  hemodynamic congestion.  Thus  if  you leave  this  part  in  the  present  paper , I recommend  to correlate  NT-proBNP with    Te etc.   Moreover, this  correlation could  be  supported in each subgroup (HFREF, HFPEF and  HFmrEF).

 Reply: this correlation has already been described in our previous stidies.

J Clin Med. 2020 Jun 16;9(6):1879. doi: 10.3390/jcm9061879.

J Cardiovasc Dev Dis. 2023 Mar 15;10(3):125. doi: 10.3390/jcdd10030125.

Biomedicines. 2022 Sep 26;10(10):2407. doi: 10.3390/biomedicines10102407.

10)   Discussion section should  be  corrected accorded to the  Results  section.

 Reply: At the present, there are no substantial correction to do, as results have not been changed

11)    Presented data was derived from  in-hospital course and  couldn’t be  translated into outpatients. Should  be  mentioned  in  limitations

 Reply: thank you for you suggestion. We have added this point among the study’s limitations.

12)     Abstract  should  be  corrected  after changes  in the  main body of  the  paper

Reply: no substantial changes have been made.

Round 2

Reviewer 2 Report

Comments and Suggestions for Authors

The  paper can be  recommended to the  Journal in current version